# Distilling LLMs' Decomposition Abilities into Compact Language Models

**Denis Tarasov** [1]   **Kumar Shridhar** [1]

## Abstract

Large Language Models (LLMs) have demonstrated proficiency in their reasoning abilities, yet their large size presents scalability challenges and limits any further customization. In contrast, compact models offer customized training but often fall short in solving complex reasoning tasks. This study focuses on distilling the LLMs' decomposition skills into compact models using offline reinforcement learning. We leverage the advancements in the LLM's capabilities to provide feedback and generate a specialized task-specific dataset for training compact models. The development of an AI-generated dataset and the establishment of baselines constitute the primary contributions of our work, underscoring the potential of compact models in replicating complex problem-solving skills[1].

## 1. Introduction

Recent strides in Natural Language Processing (NLP) have brought forth powerful Large Language Models (LLMs) like GPT-4 (OpenAI, 2023), Claude 2[2], or Gemini (Team et al., 2023). These models not only excel at straightforward tasks such as summarization and sentiment analysis but, with adept prompting, demonstrate proficiency in handling reasoning tasks that demand mathematical and logical abilities (Huang & Chang, 2022). Notably, Chain-of-Thoughts (CoT) prompting (Wei et al., 2022) and its variations (Kojima et al., 2022; Wang et al., 2022) have proven to be promising and relatively simple techniques for enhancing LLMs' reasoning capabilities.

Within the realm of complex reasoning, the ability to decompose intricate questions into a set of simpler sub-questions represents a crucial and understudied component (Shridhar et al., 2022). While existing works predominantly focus on end-to-end solutions for reasoning (Zhou et al., 2022; Lyu et al., 2023), the specific aspect of breaking down complex questions into simpler components has received limited attention. The creation of specialized datasets and benchmarks is integral to advancing the field of Deep Learning (Guss et al., 2019; Vinyals et al., 2019; Fu et al., 2020; Kurenkov et al., 2023). This work addresses the gap in understanding and exploration of the reasoning sub-questioning process by providing a dataset and baselines for further research in this direction.

Compounding the challenge is the computational overhead associated with large model sizes, making reasoning tasks computationally expensive and time-consuming when tuning models. Concurrently, approaches similar to Chain-of-Thoughts (CoT) may incur expenses, given that models with superior reasoning abilities are not available for free. In response, distilling distinct components of the reasoning process into smaller models emerges as a promising avenue for research. Decomposition, particularly in the context of teaching smaller models, proves advantageous due to their cost-effectiveness, reduced computational requirements, and accessibility.

Reinforcement Learning (RL) has demonstrated remarkable success across various domains with recent success in the NLP domain (Bai et al., 2022a;b; OpenAI, 2023; Team et al., 2023). However, some of the most popular approaches like Reinforcement Learning with Human Feedback (RLHF) (Ouyang et al., 2022) demand substantial data, and online approaches require extensive interactions with the environment. Offline RL (Levine et al., 2020), an alternative that utilizes data directly, holds potential with limited dataset sizes. This subfield has recently witnessed a surge in development, leading to a growth of diverse approaches (Kumar et al., 2020; Fujimoto & Gu, 2021; Kostrikov et al., 2021; An et al., 2021; Akimov et al., 2022; Yang et al., 2022; Ghasemipour et al., 2022; Nikulin et al., 2023). Numerous successful applications of offline RL exist in fields like robotics (Smith et al., 2022; Kumar et al., 2021), autonomous driving (Diehl et al., 2021), recommendation systems(Chen et al., 2022) and even drug-design (Tarasov et al., 2023b). For the NLP domain, recent studies have intriguingly revealed that AI feedback closely resembles human

---

[1]Department of Computer Science, ETH Zürich, Zürich, Switzerland. Correspondence to: Denis Tarasov <tarasovd@ethz.ch>.

*AI for Math Workshop at 41ˢᵗ International Conference on Machine Learning*, Vienna, Austria. Copyright 2024 by the author(s).

[1]Our code and dataset are available at https://github.com/DT6A/GSM8K-AI-SubQ

[2]https://www.anthropic.com/index/claude-2

feedback (Lee et al., 2023), and language models can be fine-tuned using their own generated feedback (Bai et al., 2022b).

In this preliminary work, we combine the advantages of sub-questioning with offline RL with feedback for the task of mathematical reasoning and distill the subquestion decomposition abilities in smaller models. Our work proposes the following: 1) an AI-generated benchmark where math questions are broken down into simpler sub-questions based on the GSM8K (Cobbe et al., 2021) dataset, 2) train smaller language models for the same task using fine-tuning and offline RL techniques to provide baselines for the task, and 3) explore the potential benefits of using AI-generated feedback on its own responses in enhancing model performance. Our experimental results reveal a big gap between Chat-GPT's reasoning abilities and what can be achieved with smaller models and existing algorithmical approaches.

## 2. Related Work

### 2.1. LM Distillation

Distillation has emerged as a pivotal technique in mitigating the computational challenges associated with LLMs while retaining their valuable knowledge. Notably, Hinton et al. (2015) introduced knowledge distillation as a means to transfer the knowledge from a complex model to a simpler one, enabling the latter to approximate the former's performance. In the context of language models, Sanh et al. (2019) successfully distilled BERT, a prominent LLM, into a more compact version named DistilBERT, maintaining competitive performance with significantly fewer parameters. Recently, a lot of work has successfully distilled the reasoning capabilities into smaller models (Yuan et al., 2023; Magister et al., 2023; Shridhar et al., 2023; Hsieh et al., 2023)

### 2.2. LLMs Reasoning

The exploration of reasoning capabilities in Large Language Models (LLMs) has been a focal point in recent NLP research. Wei et al. (2022) introduced Chain-of-Thoughts (CoT), a method compelling LLMs to approach problems in a step-by-step manner rather than providing direct answers. Noteworthy is their revelation that reasoning abilities manifest prominently in larger LM sizes. This technique involves prompting the model with step-by-step problem-solving and furnishing multiple examples for guidance through few-shot prompting. Building on this, Kojima et al. (2022) demonstrated that reasoning abilities may emerge even without examples, utilizing zero-shot prompting. Further refinement in reasoning quality was achieved by Wang et al. (2022), who showed that applying CoT multiple times and picking the most frequent answer boosts resulting performance.

### 2.3. Sub-Questioning

The paradigm of splitting reasoning into sub-question generation and sub-question answering has proven beneficial in solving reasoning problems (Shridhar et al., 2022). Notably, Shridhar et al. (2023) took a step further by distilling LLMs reasoning abilities into substantially smaller models through finetuning, laying the groundwork for our current work. It is essential to note that both of these works primarily address the entire reasoning process rather than dissecting its individual components. In a parallel thread of research, Zhou et al. (2022) demonstrated that decomposing reasoning problems into sub-problems results in improved LLM performance. Their findings indicate that LLMs can execute this split step-by-step, outperforming CoT-based approaches.

Extending beyond mere decomposition, Juneja et al. (2023) introduced a finetuning approach where a "helper" LM poses questions to a "solver" LM, guiding it toward the correct solution based on the problem and the sequence of prior interactions. Despite showcasing the efficacy of this approach, its reliance on an online RL paradigm and the use of large 13 or 33 billion parameters LLaMA models (Touvron et al., 2023) poses practical challenges. The inherent computational demands and potential instability of online RL, compounded by the necessity for significant interactions and potential cost implications, underscore the complexities associated with this method.

## 3. Preliminaries

### 3.1. Language Modeling

Our study leverages the preeminence of attention-based Transformer architectures, as introduced by Vaswani et al. (2017). In the realm of autoregressive models, we turn our attention to the widely adopted GPT-2 architecture (Radford et al., 2019), specifically selecting models of various sizes to tailor our experiments. The chosen finetuning strategies center around the application of standard cross-entropy loss, optimizing the models for accurate next-token prediction. For every approach we provide mathematical problem as a prefix and expect model to generate the sub-questions required to solve the problem.

### 3.2. Offline RL task formulation

Given the recent advancements in offline RL, we incorporate a baseline from this field. Building upon the formulation proposed by Snell et al. (2022), we cast the text generation problem as a token-level Partially Observable Markov Decision Process (POMDP). In this framework, the agent's observations correspond to prefixes of tokens, and the agent's action pertains to the selection of the next token to be generated.

# 4. GSM8K-AI-SubQ Dataset

In the pursuit of advancing research in reasoning within Language Models (LMs), we introduce the GSM8K-AI-SubQ dataset, uniquely designed to emphasize sub-questioning and leverage AI-generated feedback for these sub-questions. Our inspiration comes from the works of Bai et al. (2022b) and Lee et al. (2023). The first work provide compelling evidence that such feedback can serve as a valuable signal for fine-tuning models. The second work shows that AI-generated data exhibits comparable quality to human-annotated data in the context of preferences. We hope that our dataset will provide any insights whether similar properties hold for the reasoning.

While acknowledging that AI models may not offer the ideal source of reasoning data, we posit the substantial benefits of such a dataset. Firstly, it serves as a valuable resource for distilling reasoning abilities into smaller models, aligning with the growing trend of efficiency in language model architectures. Importantly, even if the data exhibits suboptimal characteristics, we draw inspiration from the success of offline RL in other fields, where it has demonstrated an ability to outperform the policy responsible for dataset collection. This resilience to suboptimal data quality mitigates concerns and underscores the dataset's potential impact on advancing reasoning capabilities. Lastly, the expedited and cost-effective nature of this data acquisition method democratizes its accessibility, allowing researchers to extend our dataset if required.

For the sake of completeness, we furnish LLM responses for the generated sub-questions, although our primary focus in this work centers on the sub-questioning. This dataset not only facilitates advancements in LM reasoning but also lays the groundwork for innovative approaches in the broader landscape of language model research.

## 4.1. Dataset Collection Process

In our dataset collection process, we leverage the capabilities of ChatGPT, specifically utilizing the `gpt-3.5-turbo-0613`[3] version. This version is chosen due to its lenient restrictions on the number of queries and its cost-effectiveness, offering responses of commendable quality when benchmarked against similar LLMs (Tunstall et al., 2023). Interactions with ChatGPT are facilitated through the OpenAI API[4], employing a temperature setting of 0.7 to ensure a diverse range of responses.

For the specific task at hand, we curate a dataset of mathematical problems extracted from the GSM8K dataset (Cobbe et al., 2021). This dataset comprises a diverse array of grade

---

[3]The most recent `gpt-3.5-turbo` version
[4]https://platform.openai.com/docs/overview

school math word problems meticulously crafted by human problem writers. While designed to be solvable by a middle school student, these problems present a noteworthy challenge for LMs. Maintaining the integrity of the original train/test split, our dataset consists of 7473 training examples and 1319 testing examples.

**Sub-questions Generation.** The initial phase of our data generation involves creation of set of sub-questions for each problem within the GSM8K dataset. We prompt ChatGPT to decompose the given problem into distinct sub-problems and provide corresponding sub-questions, aiming to elucidate the problem-solving process for others. To guide the model and ensure a consistent output format, we furnish two examples of problems and their corresponding sub-questions as demonstration inputs. Each request is treated as an independent dialogue to eliminate potential interference. We keep a 2-shot prompting strategy and preserving independence for all subsequent queries. A sample input and output for this sub-question generation process are illustrated in Table 10. To enhance the dataset's size and diversity, we repeat the inquiry for each problem three times, resulting in 22,419 training samples. While the repetition could be further increased for creating a larger dataset, we acknowledge budget constraints as a limiting factor in this study.

**Answers Generation.** In the subsequent phase of our data collection, we focused on generating responses to the previously obtained sub-questions. Tasking ChatGPT solely with the responsibility of providing answers to these queries and obtaining a final answer for the original problem. An illustrative example of this prompt-response interaction is presented in Table 11. The generated final answers were then employed as labels to categorize each set of sub-questions. Sub-questions associated with an original problem solution were identified as "good", while those failing to lead to a resolution were categorized as "bad".

**Feedback Generation.** Concluding our dataset collection, we engage in the pivotal task of soliciting feedback from ChatGPT on its generated sub-questions. The objective is to introduce a nuanced signal at the individual question level, mitigating sparsity concerns associated with relying solely on the correctness of final answers. This aspect assumes significance, especially in the context of potential RL applications. Drawing inspiration from the effective sub-questioning strategy proposed by Shridhar et al. (2022), we task ChatGPT with determining the usefulness of each sub-question in the problem-solving process. Refer to Table 12 for an exemplary interaction. To account for potential inconsistencies in ChatGPT's feedback, we query the model three times for each set of sub-questions. This repetition strategy aligns with findings in recent studies (Wang et al., 2022; Lee et al., 2023), showing that leveraging multiple

Table 1: Train set statistics for problems splitted by the number of sub-question sets which lead to the correct answer. "Problem length" denotes number of characters.

| METRIC | 0 CORRECT | 1 CORRECT | 2 CORRECT | 3 CORRECT |
|---|---|---|---|---|
| # PROBLEMS | 1343 (18%) | 866 (11%) | 1139 (15%) | 4052 (54%) |
| AVG LENGTH | 269.4 ± 106.7 | 252.1 ± 100.3 | 240.2 ± 94.7 | 217.5 ± 82.9 |
| MEDIAN LENGTH | 250 | 235.5 | 225 | 201.5 |

responses often results in more useful data. It's worth noting that the repetition factor can be adjusted for further data refinement. Subsequently, scores for each sub-question are computed by evaluating the fraction of responses deeming it useful, establishing a metric for assessing the efficacy of individual sub-questions.

**Dataset Collection Costs.** The compilation of our training set demanded approximately 12 hours of real-time investment and incurred a cost of approximately $100 for utilizing the `gpt-3.5-turbo` version. Notably, feedback generation constituted around 70% of the total costs, primarily attributed to the necessity of acquiring feedback multiple times for each set of questions.

### 4.2. Dataset Analysis

In this section, we present an analysis of the collected training data, offering valuable statistics and insights.

The distribution of sub-question set sizes is visualized in Figure 5. While the majority of sets comprise 2 to 6 questions, some outliers exist, featuring either a single question or more than 6. Notably, we opt to retain these outliers in our dataset for a more diverse representation.

Our analysis extends to evaluating the efficacy of sub-question sets in leading to the correct solution for each problem. As outlined in Table 1, approximately 54% of the sets consistently resulted in a correct solution across all three attempts, while only around 18% failed to yield a correct solution in any instance. Moreover, longer problems appear to pose increased difficulty, aligning intuitively with expectations. Calculating average accuracy by treating each sub-question set as an independent problem, the overall accuracy for the training set stands at 0.68.

To delve deeper into the relationship between problem complexity and sub-question count, we compare distributions for 0, 1, and 2 out of 3 correct answers versus 3 out of 3 correct answers in Figure 2, Figure 3, Figure 4. The analysis reveals a discernible trend: easier problems tend to be associated with a smaller number of sub-questions.

Continuing our exploration, we investigate whether Chat-GPT's feedback on its own sub-questions holds intrinsic value and diverges from random noise.

The distribution of usefulness scores for each sub-question, depicted in Figure 6, indicates that approximately 90% of sub-questions received consistent markings as useful across all three iterations. However, to gain a more nuanced perspective, we average the usefulness scores for each sub-question set and showcase the distribution of averaged confidences in Figure 7.

Upon examining the fraction of problems with at least one negative response, we find that 36% of problems fall into this category, aligning with the 54% that were correctly solved 3 out of 3 times. Recognizing that a single misstep in the reasoning process can lead to an incorrect answer, we employ the presence of negative feedback as an indicator for predicting an incorrect solution. However, this indicator comes with limitations, such as the potential for false negatives arising from the model providing the correct final answer despite a flawed reasoning step, and the occasional presence of randomness during feedback collection. The confusion matrix in Figure 8 outlines this setup, showcasing differences from a random scenario. Notably, when negative feedback is absent, 72% of problems were actually solved correctly, compared to 62% when negative feedback is present.

To quantitatively assess the performance of this heuristic, we compute the ROC AUC score, treating the average sub-question set usefulness as a probability and designating an incorrect answer as the "target label". The resulting ROC AUC score of 0.56 indicates a departure from randomness (0.5). Further, the Pearson correlation coefficient, calculated at -0.09 with a p-value of $10^{-46}$, suggests that a higher usefulness score corresponds to a decreased likelihood of encountering an incorrect answer. Despite the simplicity of this heuristic, it provides informative cues on the efficacy of the sub-question set, indicating its potential utility.

## 5. Baselines

In our exploration of the constructed dataset, we conduct a series of experiments utilizing both supervised techniques and an offline RL approach. Our experiments involve the deployment of pretrained small and medium versions of GPT-2 (Radford et al., 2019) as well as pretrained DistilGPT (Sanh et al., 2019) to gauge performance dynamics across various models sizes. The GPT-2 medium model, with 345 million parameters, represents the largest model in our experiments. This choice is deliberate, as larger models endowed with reasoning abilities often exceed 7 billion parameters, rendering their usage complex for many researchers. Unfortunately, due to computational resource limitations, our experiments did not extend to GPT-2 large and XL models.

## 5.1. Evaluation Protocol

To facilitate a robust comparison of different approaches, we employ the same version of ChatGPT utilized during data collection. Sub-questions generated by each approach are presented to ChatGPT using the same prompt from Table 11, and we measure the accuracy of final answers as our evaluation metric. Standardizing the evaluation environment, we set the temperature to 0 and fix the random seed to ensure reproducibility and eliminate randomness during the evaluation process. This rigorous evaluation protocol ensures a fair and consistent assessment of the performance across various approaches. One evaluation of test set costed approximately 1.5$ with `gpt-3.5-turbo`.

We additionally use the same protocol to evaluate results using open-source models Mistral 7B (Jiang et al., 2023), LLaMA 7B and LLaMA 13B (Touvron et al., 2023).

## 5.2. Applied Approaches

In this subsection, we provide a brief overview of each applied approach. For a more details, refer to Appendix A.

**Behavioral Cloning.** Behavioral Cloning (BC) is a fundamental and robust approach commonly applied to datasets involving decision-making tasks. In the context of NLP, it translates to fine-tuning a language model to replicate a specific behavior or policy observed in the dataset. To select the best model, we employ a small held-out fraction (1%) of the training data. Following the methodology inspired by Shridhar et al. (2022), we use the BLEU score (Papineni et al., 2002) calculated between the generated questions and those produced by ChatGPT. Given that the primary goal of BC is to replicate the original policy's behavior, BLEU serves as a suitable metric, indicating the similarity between two texts. Our empirical observation show that BLEU correlates with the final performance, making it a reasonable choice for model evaluation in the context of BC. For all subsequent approaches, the best BC model serves as the initialization for the LM.

**Filtered Behavioral Cloning.** Filtered BC (Chen et al., 2021) introduces a modification of BC by considering only a fraction of the best trajectories in the dataset. This approach proves particularly effective when a substantial number of high-quality examples are at disposal. In the context of our task, we exclusively retain samples corresponding to sub-question sets that result in the correct solution. The model selection process remains consistent with the standard BC approach.

**Implicit Language Q-Learning.** Implicit Language Q-Learning (ILQL) (Snell et al., 2022) represents an adaptation of the offline RL approach known as IQL (Kostrikov et al., 2021) to NLP tasks. The core idea behind ILQL involves training additional Value (V) and Q-function heads with IQL

objectives. These additional functions are then employed to reweight the original LM outputs using the advantage value, which is the difference between V and Q values.

The selection of ILQL is motivated by the effectiveness of IQL as one of the strongest offline RL approaches in diverse domains (Tarasov et al., 2022). Given the limited adaptation of offline RL approaches to NLP problems, ILQL emerges as the state-of-the-art choice. Given that IQL optimizes for rewards, which may not inherently correlate with the dataset policy, selecting the best model becomes challenging. In the absence of a clear best model selection criterion, we have tried to use the same criterion as we did for BC and the common offline RL practice of taking the latest checkpoint after training. The first method produced slightly better results on average.

Two versions of ILQL are tested in our experiments: ILQL-full utilizing all available rewards from the dataset and ILQL-sparse employing only answer correctness as rewards.

## 5.3. Experimental Results

The experimental results, summarized in Table 2, offer insights into the performance of the proposed approaches averaged over various answering LLMs. For a detailed breakdown based on specific answering models, refer to Appendix B.

| Algorithm | DistillGPT | GPT-2 small | GPT-2 medium | Average |
|---|---|---|---|---|
| BC | 0.255 | 0.284 | 0.310 | 0.283 |
| Filtered BC | **0.260** | **0.293** | **0.319** | **0.291** |
| ILQL-sparse | 0.249 | 0.278 | 0.308 | 0.278 |
| ILQL-full | 0.256 | 0.277 | 0.306 | 0.280 |
| ChatGPT | N/A | N/A | N/A | 0.429 |

Table 2: Accuracy score of the final answer avereged over different models which were used for sub-questions answering. Best scores are highlighted with **bold**.

It is evident that all tested approaches fall short when compared to the ChatGPT, indicating substantial room for improvement. Sub-questions generation abilities also improve with the size of the backbone model which alligns with previous researches.

Filtered BC demonstrates improved performance over standard BC in most scenarios, consistent with expectations when sufficient amount of high-quality demonstrations are present in the dataset. However, this trend is not universal, particularly when LLaMAs serve for question answering and GPT-2 medium is a backbone model for sub-question generation. Interestingly, Filtered BC outperforms BC only when GPT-2 medium is used alongside Mistral for answering.

The comparison between ILQL-sparse and ILQL-full does not reveal a consistent advantage for either method. In most of the cases, both ILQL variants underperform Filtered BC,

and even falling behind standard BC in many of the cases. However, the superiority of the Filtered BC over offline RL approaches in NLP was also recently demonstrated by Gulcehre et al. (2023). Our finding should solve as additional motivation for the development of offline RL algorithms for NLP.

## 6. Limitations and Future Work.

Our work serves as a foundational exploration, opening avenues for various future directions.

**Development of Offline RL Approaches:** A pivotal area for future exploration involves advancing offline RL or other suitable methodologies for distilling reasoning abilities from static datasets. This extension could contribute to more effective utilization of language models in reasoning tasks.

**Creation of a Larger Benchmark:** Expanding our methodology, future work could focus on generating a more extensive benchmark as it requires only the access to ground truth-answers in the datset which usualy holds. This benchmark might incorporate a diverse set of reasoning datasets, such as MATH (Hendrycks et al., 2021) or AQuA (Ling et al., 2017), providing a broader assessment of reasoning capabilities.

**Concentration on Sub-Question Answering:** Delving deeper into the sub-question answering aspect of the reasoning process presents a promising direction. While our dataset includes ChatGPT responses for sub-questions, their scoring and utilization remain unexplored. Future studies could investigate this component to enhance understanding and performance.

**Utilization of Open-Source Models:** Exploring the application of open-source models, such as LLaMA, for sub-question generation emerges as a cost-effective alternative. Accessible without financial constraints, these models present an opportunity for researchers to delve into sub-question generation without monetary limitations. We were not able to run such kind of experiments ourselves due to the computational limitations.

## 7. Conclusion

This work introduces a novel AI-generated benchmark tailored for evaluating sub-questioning in reasoning tasks. We employ diverse offline learning approaches, varying model sizes for baselines, and assess the performance using different LLMs. Our experiments aim to shed light on the challenges and potential avenues for enhancing reasoning capabilities.

The outcomes reveal a significant performance gap between the best-performing approach and ChatGPT. The under-whelming performance of the offline RL approach underscores the need for further advancements in this domain, presenting an opportunity for future research to explore and refine these methodologies.

By providing this benchmark, we aspire to catalyze research endeavors in the realm of sub-questioning. We anticipate that the dataset curated in this work will serve as a foundational resource for assessing the reasoning capabilities of emerging offline RL approaches in the field of NLP.

## 8. Acknowledgements

We thank Mrinmaya Sachan for his supervision of our work and Elliott Ash for providing us with computational resources.

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

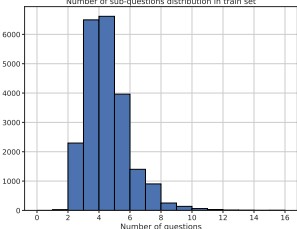

Figure 1

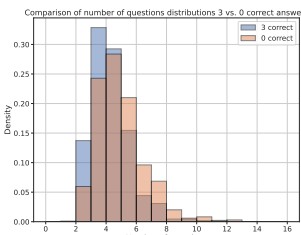

Figure 2

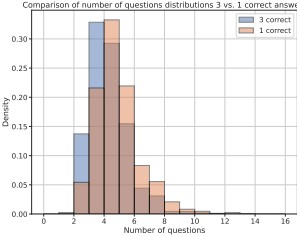

Figure 3

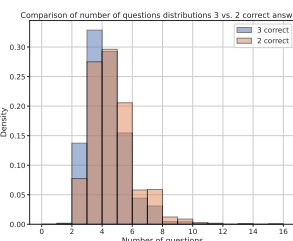

Figure 4

Figure 5: Number of questions distributions in the train set. Figure 1 Distribution in the entire train set with mean 4.0 ± 1.5 and median 4, (Figure 2, Figure 3, Figure 4) Comparisons of distributions in the number of questions between problems with 3 vs. 0, 1, 2 sets of sub-questions that lead to the correct final answer.

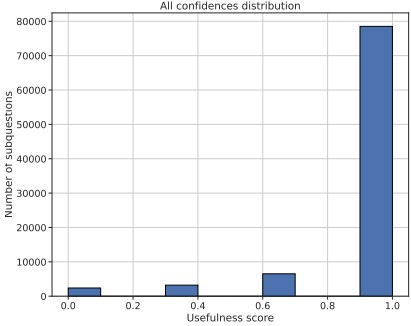

Figure 6

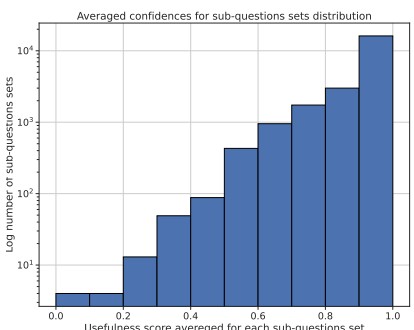

Figure 7

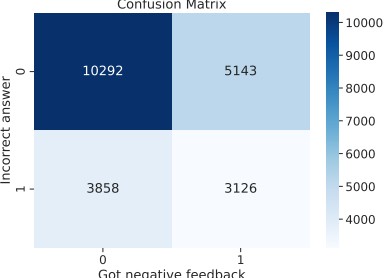

Figure 8

Figure 9: (a) distribution of the usefulness feedback over individual sub-questions, (b) distribution of the usefulness feedback averaged over sets of sub-questions, (c) confusion matrix between the presence of negative feedback in the set of sub-questions and correctness of the final answer based on corresponding sub-questions.

## A. Experimental Details

We run all of our experiments using single V100 GPUs. The training time never exceeded 5 days.

Hyperparaters are kept the same across different model sizes.

### A.1. Hyperparameters

We conducted hyperparameters search only through $\{0.5, 0.7, 0.9\}$ values for IQL $\tau$ and $\{0.5, 1.0, 3.0\}$ values for ILQL $\beta$ parameter using GPT-2 small with full reward. Also, following Tarasov et al. (2023a) we have increased discount factor value from default 0.99 to 0.999 which improved ILQL performance.

| Hyperparameter | Value |
|---|---|
| Batch size | 32 |
| Optimizer | Adam |
| Learning rate | 1e-4 |
| Gradient steps | 10000 |

Table 3: BC hyperparameters.

| Hyperparameter | Value |
|---|---|
| Batch size | 32 |
| Optimizer | Adam |
| Learning rate | 1e-4 |
| Gradient steps | 7500 |

Table 4: Filtered BC hyperparameters.

| Hyperparameter | Value |
|---|---|
| Batch size | 32 |
| Optimizer | Adam |
| Learning rate | 1e-4 |
| Gradient steps | 25000 |
| Discount factor | 0.999 |
| Target update rate | 5e-3 |
| IQL $\tau$ | 0.9 |
| ILQL $\beta$ | 1.0 |
| V loss weight | 1.0 |
| Q loss weight | 1.0 |
| CQL loss weight | 0.01 |

Table 5: ILQL hyperparameters.

## B. Full Tabular Scores

| Algorithm | DistillGPT | GPT-2 small | GPT-2 medium | Average |
|---|---|---|---|---|
| BC | 0.476 | 0.508 | 0.538 | 0.507 |
| Filtered BC | **0.493** | **0.527** | **0.576** | **0.532** |
| ILQL-sparse | 0.474 | 0.513 | 0.531 | 0.506 |
| ILQL-full | 0.482 | 0.505 | 0.533 | 0.507 |
| ChatGPT | N/A | N/A | N/A | 0.682 |

Table 6: Accuracy score of the final answer using ChatGPT for sub-questions answering. Best scores are highlighted with **bold**.

| Algorithm | DistillGPT | GPT-2 small | GPT-2 medium | Average |
|---|---|---|---|---|
| BC | 0.118 | 0.154 | **0.164** | 0.145 |
| Filtered BC | **0.125** | **0.159** | 0.162 | **0.149** |
| ILQL-sparse | 0.122 | 0.141 | **0.164** | 0.142 |
| ILQL-full | 0.123 | 0.147 | 0.163 | 0.144 |
| ChatGPT | N/A | N/A | N/A | 0.234 |

Table 7: Accuracy score of the final answer using LLaMA 7B for sub-questions answering. Best scores are highlighted with **bold**.

| Algorithm | DistillGPT | GPT-2 small | GPT-2 medium | Average |
|---|---|---|---|---|
| BC | 0.184 | 0.212 | **0.247** | 0.214 |
| Filtered BC | **0.194** | **0.230** | 0.245 | **0.223** |
| ILQL-sparse | 0.178 | 0.204 | **0.247** | 0.210 |
| ILQL-full | 0.183 | 0.205 | **0.247** | 0.212 |
| ChatGPT | N/A | N/A | N/A | 0.353 |

Table 8: Accuracy score of the final answer using LLaMA 13B for sub-questions answering. Best scores are highlighted with **bold**.

| Algorithm | DistillGPT | GPT-2 small | GPT-2 medium | Average |
|---|---|---|---|---|
| BC | **0.240** | **0.264** | 0.290 | **0.265** |
| Filtered BC | 0.228 | 0.256 | **0.293** | 0.259 |
| ILQL-sparse | 0.223 | 0.253 | 0.288 | 0.255 |
| ILQL-full | 0.235 | 0.252 | 0.282 | 0.256 |
| ChatGPT | N/A | N/A | N/A | 0.446 |

Table 9: Accuracy score of the final answer using Mistral for sub-questions answering. Best scores are highlighted with **bold**.

## C. ChatGPT Prompts

| **Input** |
| --- |
| You are given mathematical problems marked with "Problem". Your task is to split it into smaller sub-problems and formulate them as sub-questions which will be answered by someone else who's objective is to solve the original problem. Questions must not contain the answers for the previous questions in them. Do not ask questions where the answer is already given in the problem. For each problem come up with the sequence of sub-questions and output each of them on separate line which starts with letter Q followed by the number of question. Do not output anything else. |
| |
| Problem: A robe takes 2 bolts of blue fiber and half that much white fiber. How many bolts in total does it take? |
| |
| Q1: How many bolts of white fiber does it take? |
| Q2: How many bolts in total does it take? |
| |
| Problem: Betty is saving money for a new wallet which costs $100. Betty has only half of the money she needs. Her parents decided to give her $15 for that purpose, and her grandparents twice as much as her parents. How much more money does Betty need to buy the wallet? |
| |
| Q1: How much did the house cost? |
| Q2: How much did the repairs increase the value of the house? |
| Q3: What is the new value of the house? |
| Q4: How much profit did he make? |
| |
| Problem: Betty is saving money for a new wallet which costs $100. Betty has only half of the money she needs. Her parents decided to give her $15 for that purpose, and her grandparents twice as much as her parents. How much more money does Betty need to buy the wallet? |
| **Response** |
| Q1: How much money does Betty have? |
| Q2: How much money did Betty's parents give her? |
| Q3: How much money did Betty's grandparents give her? |
| Q4: How much money does Betty still need to buy the wallet? |

Table 10: Sample ChatGPT input for generating sub-questions and one of the corresponding responses.

---

**Input**

---

You are given the mathematical problems marked with "Problem" and a sequence of questions which should help in solving it. Question number N is marked as "QN:". Based on the problem and sequence of questions answer each of the questions with answer "AN:" and give the answer to the whole problem using "Final answer:" using only the resulting number without adding any additional comments after it.

Problem: A robe takes 2 bolts of blue fiber and half that much white fiber. How many bolts in total does it take?

Q1: How many bolts of white fiber does it take?
Q2: How many bolts in total does it take?

A1: It takes 2/2=1 bolt of white fiber
A2: So the total amount of fabric is 2+1=3

Final answer: 3

Problem: Josh decides to try flipping a house. He buys a house for $80,000 and then puts in $50,000 in repairs. This increased the value of the house by 150%. How much profit did he make?

Q1: How much did the house cost?
Q2: How much did the repairs increase the value of the house?
Q3: What is the new value of the house?
Q4: How much profit did he make?

A1: The cost of the house and repairs came out to 80,000+50,000=130,000
A2: He increased the value of the house by 80,000*1.5=120,000
A3: So the new value of the house is 120,000+80,000=200,000
A4: So he made a profit of 200,000-130,000=70,000

Final answer: 70000

Problem: Betty is saving money for a new wallet which costs $100. Betty has only half of the money she needs. Her parents decided to give her $15 for that purpose, and her grandparents twice as much as her parents. How much more money does Betty need to buy the wallet?

Q1: How much money does Betty have?
Q2: How much money did Betty's parents give her?
Q3: How much money did Betty's grandparents give her?
Q4: How much money does Betty still need to buy the wallet?

---

**Response**

---

A1: Betty has 100/2=50 dollars.
A2: Betty's parents gave her 15 dollars.
A3: Betty's grandparents gave her 15*2=30 dollars.
A4: Betty still needs 100-50-15-30=5 dollars.

Final answer: 5

---

Table 11: Sample ChatGPT input for answering sub-questions and the corresponding response.

| Input |
| --- |
| You are given the mathematical problem marked with "Problem" and a sequence of sub-questions for solving it. Sub-question number N is marked as "QN:". Based on the problem for each sub-questions decide whether this question is helpful for solving the given problem. An essential property of a good questioning strategy is to ask questions that are directed towards the most critical domain specific content. Asking the right sequence of relevant questions that can assist in reaching the final goal is an important part of good questioning. If question repeats any of the previous it is not useful. The question for which answer is given in the problem or can't be answered at all is also not useful. So redundant questions are not good. |
| For each question output me "QN: <Yes/No>" and only it where N is the number of the question, e.g. "Q1: <Yes/No> Q2: <Yes/No> for the first two questions. Do not try to solve the problem anyhow as I'm only interested in the quality of the sub-questions. Strictly follow the output format. Provide answers only for the last given problem. |
| Problem: Janet's ducks lay 16 eggs per day. She eats three for breakfast every morning and bakes muffins for her friends every day with four. She sells the remainder at the farmers' market daily for $2 per fresh duck egg. How much in dollars does she make every day at the farmers' market? |
| Q1: How many eggs does Janet sell? 
 Q2: Is duck an animal? 
 Q3: How many eggs does each duck lay? 
 Q4: How much does Janet make at the farmers' market? |
| Q1: Yes 
 Q2: No 
 Q3: No 
 Q4: Yes |
| Problem: A robe takes 2 bolts of blue fiber and half that much white fiber. How many bolts in total does it take? |
| Q1: How many bolts of white fiber does it take? 
 Q2: How bolts of blue fiber does it take? 
 Q3: How bolts of white fiber does it take? 
 Q4: How many bolts in total does it take? |
| Q1: Yes 
 Q2: No 
 Q3: No 
 Q4: Yes |
| Problem: Betty is saving money for a new wallet which costs $100. Betty has only half of the money she needs. Her parents decided to give her $15 for that purpose, and her grandparents twice as much as her parents. How much more money does Betty need to buy the wallet? |
| Q1: How much money does Betty have? 
 Q2: How much money did Betty's parents give her? 
 Q3: How much money did Betty's grandparents give her? 
 Q4: How much money does Betty still need to buy the wallet? |
| **Response** |
| Q1: Yes 
 Q2: Yes 
 Q3: Yes 
 Q4: Yes |

Table 12: Sample ChatGPT input for generating sub-questions feedback and one of the corresponding responses.

