# OpenReview forum: "Distilling LLMs’ Decomposition Abilities into Compact Language Models"
_ICML.cc/2024/Workshop/AI4MATH — ICML 2024 Workshop AI4MATH Poster_

### Official Review · Reviewer_ESCA · 2024-06-11

**Rating:** 8
**Confidence:** 5

**Summary:**

The paper focuses on transferring the decomposition skills of Large Language Models (LLMs) into smaller, more compact models using offline reinforcement learning (RL). This approach aims to address the scalability challenges and high computational costs associated with LLMs while leveraging their reasoning abilities.

**Questions:**

just curious about computational cost, what is the computational cost associated with training the compact models using offline RL compared to traditional methods?

**Reasons To Accept:**

Originality:
The paper introduces a novel method for distilling the decomposition skills of LLMs into smaller models using offline RL, addressing a relatively unexplored area in NLP. The creation of an AI-generated dataset focused on sub-question generation for reasoning tasks is innovative and valuable for future research.

Quality:
The methodology is well-developed, leveraging recent advances in reinforcement learning and language model fine-tuning. The experiments are thorough, involving multiple baselines and different model sizes, providing a robust validation of the proposed approach.

Clarity:
The paper is well-organized, with clear explanations of the dataset generation process, experimental setup, and results. The use of tables and figures helps in understanding the distribution and effectiveness of the generated sub-questions.

Significance:
The ability to distill complex reasoning skills into smaller models has significant implications for making advanced NLP capabilities more accessible and cost-effective. The introduced dataset and baselines provide a foundation for future research in sub-question generation and offline RL in NLP.

**Reasons To Reject:**

n/a

---

### Official Review · Reviewer_UKBB · 2024-06-12

**Rating:** 7
**Confidence:** 4

**Summary:**

This paper focuses on distilling question decomposition abilities from LLMs into small PLMs. It constructed a dataset for this task by decomposing questions from GSM8K using GPT-3.5. And it provides several baselines on its released benchmark.

**Questions:**

1. I am interested in the relationship between the question difficulty and the number of decomposed sub-questions. In your paper, you measured “question difficulty” by “question length”. I prefer the measure by “the number of steps of the reference solution provided to the question in the dataset”.

2. It might not be suitable to claim “LLMs can execute this split step-by-step, outperforming CoT-based approaches” when introducing related works in Section 2.3. This statement is not recognized by the community and your claim is too strong. Firstly, the current mainstream is CoT-based approaches and only 2 references of decomposition-based empirical approaches might not be enough to challenge a mainstream method. Secondly, according to the intuition, decomposition-based approaches have a longer pipeline than CoT, which could lead to heavier error propagation issue, affecting its efficacy.

3. “Filtered BC outperforms BC only when GPT-2 medium is used alongside Mistral for answering”. If the results of different answering models are only presented in the appendix, I recommend this analysis also moved to the appendix.

4. You missed Section 4.2

**Reasons To Accept:**

Suggest a new direction for distilling reasoning abilities into small models and release a dataset.

**Reasons To Reject:**

1. The motivation of distilling decomposition abilities needs more defense. This affects the value of the proposed new task.
The authors provided two motivations for distilling decomposition abilities in the paragraph 2-3 of Introduction. Paragraph 2 claims there is a lack (thus it could be a novel direction). Paragraph 3 claims distilling sub-components of reasoning like decomposition could be better than distilling CoT. Here we focus on the latter motivation.

Notably, there is not any reference in paragraph 3. The authors claimed that using decomposition is cheaper than CoT. However, this claim is neither intuitive nor supported by any evidence in the paper. The last sentence claims the decomposition “proves advantageous” but don’t give a reference. Moreover, if we’re going to distill sub-components of reasoning into small models, should we consider the cost of the entire pipeline (decomposition + sub-question answering) rather than only the single component (i.e. decomposition here)?

The first paragraph of Section 2.3 “Sub-Questioning” in “Related Work” mentions some empirical success of decomposition-based approaches with LLMs. However, firstly these related works mainly focus on the effectiveness rather than the efficiency. Secondly they studied reasoning with LLMs rather than distilling reasoning into small models. So there are two gaps from them to the authors’ motivation.

As the authors propose a new paradigm of distilling reasoning abilities, I strongly recommend them to take the entire pipeline into the consideration and compare it with CoT-based distillation. The comparison of the two paradigms could be of more interest, benefit more to the community and bring more insights than set the baseline results of the single task.


2. The methodology and analysis of RL approaches are short of convincing. They need examination.
The authors used the binary feedbacks from GPT-3.5 as rewards for RL training, without quality examination. They referred to “Rlaif: Scaling reinforcement learning from human feedback with ai feedback” as evidence to claim that the quality of AI feedbacks is comparable to of human. However, there is an important gap between this evidence and the methodology in this paper: the feedback types are very different. The reference paper is pairwise-based feedback which assigned a relative comparison label to only one target each time, while this paper collected non-comparison and absolute labels for multiple targets. Moreover, I concerns whether GPT-3.5 is reliable enough for automatic evaluation.

As described above, I speculate the rewards could be noisy. So there might exist a large quality difference between SFT training data and RL rewards. In this case, it’s not robust to claim that SFT is better than ILQL-full algorithms.

Moreover, it is strange to claim that this paper focuses on ”distilling the LLMs’ decomposition skills into compact models using offline reinforcement learning” in Abstract, as the best experimental results in this paper are brought by SFT approaches.

---

### Official Review · Reviewer_CzTi · 2024-06-13

**Rating:** 4
**Confidence:** 4

**Summary:**

This paper proposes a method using LLMs to generate subquestions for math reasoning and trains smaller models for this task.

**Questions:**

Typos:
1. Citation on page 1, why do you cite the claude-2 as your code?
2. Some citation formats are wrong, for example, in section 2.1, the citations should use \citep

Questions:
1. Why do we need smaller models to do the decomposition? This task is not sensitive to latency. Correctness should be the most important thing to care about.
2. The motivation and contributions of this paper are unclear. I'm confused about whether you want to propose a dataset or a method for training smaller models. Your title seems that you want to propose a method, but the conclusion says you propose a benchmark.
3. How do you check whether the generated subquestions are correct? I didn't see any human check process in the generated dataset. Is there a probability of generating wrong or meaningless subquestions?

**Reasons To Accept:**

The paper proposes a novel benchmark for sub-questioning task.

**Reasons To Reject:**

1. The necessity of this task is unclear.
2. Only ChatGPT is used for the task, open-sourced models like LLaMA-3 are not tested.
3. The correctness of the AI-generated dataset is unchecked.

---

### Meta-Review · Area_Chair_Cgde · 2024-06-13

**Recommendation:** Accept (Poster)
**Confidence:** 4

**Metareview:**

Being able to decompose a question into subquestions is indeed an important question as a proxy of reasoning in mathematics. I essentially agree with the comments of the two first reviewers, and the main limits are: 1) it is not clear that the correctness of the dataset is properly checked, it seems to be essentially language models checking language models. 2) It would be interesting to have more motivation for the task by showing how useful it is to solve problems, and in particular why is distillation of such capacity so useful. Nevertheless, I believe that at least the second one can be fixed easily and I think the approach is interesting and worth acceptance.

---

### Decision · Program_Chairs · 2024-06-13

Accept (Poster)